materials science

sponge aerogels, pomelo peel, silanization, oil-adsorption, recyclability

**Author for correspondence:**
Fengzhi Tan
e-mail: fz_tan@126.com; tanfz@dlpu.edu.cn

This article has been edited by the Royal Society of Chemistry, including the commissioning, peer review process and editorial aspects up to the point of acceptance.

# Controllable synthesis of pomelo peel-based aerogel and its application in adsorption of oil/organic pollutants

## Guangyu Shi[1], Yizhu Qian[2], Fengzhi Tan[1], Weijie Cai[1], Yuan Li[1] and Yafeng Cao[1]

[1]School of Light Industry and Chemical Engineering, Dalian polytechnic university, No. 1, Qinggongyuan, Ganjingzi District, Dalian 116034, People's Republic of China
[2]Dalian No. 24 high school. No. 217, Jiefang Road, Zhongshan District, Dalian 116001, People's Republic of China

FT, 0000-0001-8715-2442

Oil/water separation is a field of high significance as it might efficiently resolve the contamination of industrial oily wastewater and other oil/water pollution. In this paper, an environmentally-friendly hydrophobic aerogel with high porosity and low density was successfully synthesized with renewable pomelo peels (PPs) as precursors. Typically, a series of sponge aerogels (HPSA-0, HPSA-1 and HPSA-2) were facilely prepared via high-speed dispersion, freeze-drying and silanization with methyltrimethoxysilane. Indeed, the physical properties of aerogel such as density and pore diameter could be tailored by different additives (filter paper fibre and polyvinyl alcohol). Hence, their physico-chemical properties including internal morphology and chemical structure were characterized in detail by Fourier transform infrared, Brunauer–Emmett–Teller, X-ray diffraction, scanning electron microscope, Thermal gravimetric analyzer (TG) etc. Moreover, the adsorption capacity was further determined and the results revealed that the PP-based aerogels presented excellent adsorption performance for a wide range of oil products and/or organic solvents (crude oil $49.8 \, \mathrm{g \, g^{-1}}$, soya bean oil $62.3 \, \mathrm{g \, g^{-1}}$, chloroform $71.3 \, \mathrm{g \, g^{-1}}$ etc.). The corresponding cyclic tests showed the absorption capacity decreased slightly from 94.66% to 93.82% after 10 consecutive cycles, indicating a high recyclability.

# 1. Introduction

Nowadays, with the rapid development of the chemical industry and offshore exploitation, oily wastewater has become an extremely common pollutant, which can cause serious damage to the water environment and ecological systems [1,2]. In addition, the frequent oil leakages and organic pollutants spillage during marine transportation or crude oil production are potentially catastrophic to marine environments, and also a great waste of valuable natural resources [3]. To address the issues arising from oil spills, organic pollutants and industrial oily wastewater, various techniques have been developed for oily wastewater treatment, such as *in situ* burning, biodegradation, flotation, physical adsorption etc. Among these techniques, physical adsorption by porous materials has been proved to be very promising and efficient in the removal of oil/organic pollutants [4,5]. Many porous absorbent materials have been prepared and deeply investigated like activated carbon [6], polypropylene non-woven fabric [7], sponges [8], propylene foam [9] etc. However, such materials generally suffer from low capacities/output, high cost, disbiodegradation and lower separation selectivity because of the simultaneous adsorption of oil and water etc. [10,11]. Hence, it is critical to develop a novel type of sorbent which possesses the advantages of sorption capacity, selectivity, low cost, recyclability and environmental friendliness [12,13].

Cellulose is one of the abundant raw materials in nature. As a 'young' third generation of aerogel materials, cellulose-based aerogels seem to be one of the most fascinating natural oil sorbents after the appropriate modifications [14]. To date, various kinds of natural fibres have been reported as a precursor, such as kapok fibre [15], cotton [16], corn straw [17], banana peels [18], softwood [19] etc. It was accepted that the oil sorption capability of cellulose-based aerogel was attributed to its low density, high porosity and the three-dimensional network structure. In order to achieve excellent oil/water selectivity, most of the cellulose-based aerogels were further modified via esterification, nanocoating, carbonization, silanization and so on. Wang *et al.* fabricated a kind of superhydrophobic cellulose sponge for oil/water separation using a one-pot hydrothermal technique and subsequent hydrophobic modification with dodecanethiol. The as-prepared kapok fibre showed good sorption capacity up to $4-70 \text{ g g}^{-1}$ for oil and organic solvents [15]. Li *et al.* prepared a novel coconut peat powder-based magnetic sorbent for selective oil–water separation. The product was modified by low-surface-energy octadecylamine, and exhibited high recyclability, with a loss of less than 15.32% in oil absorption capacity after 11 absorption–desorption cycles [20]. Additionally, Lu *et al.* reported a superhydrophobic magnetic ethyl cellulose (EC) sponge with low density (less than $18 \text{ mg cm}^{-3}$) and high porosity (greater than 98%) [21]. Furthermore, the sponge showed excellent separation efficiency and good absorption capacity for oils and organic solvents ($37-71 \text{ g g}^{-1}$). However, compared with natural fibre, to the best of our knowledge there has been little research on the aerogel-based citrus fruits peel.

It was reported that the global production of citrus fruits amounts to over 100 million tons since 2010 [22,23]. A large amount of citrus waste was generated, which accounted for almost 50wt% of the fruit weight. Pomelo peel (PP), as an abundant pectin-rich fruit waste of Southeastern Asia, possessed *ca* 40 wt% of pomelo. Unfortunately, most of the PP was directly discarded in landfills, which resulted in resource waste and environmental pollution [24]. It should be noted that PP contained rich plant fibre, insoluble polysaccharides (e.g. cellulose, hemicellulose, pectin) and ligin [25]. This chemical composition rendered the PP a promising material for the cellulose-based aerogel. Furthermore, as a low-cost natural absorbent, PP presented high oil sorption capacity due to its porous structure, especially macropores ($2-20 \mu m$), which might provide sufficient space to adsorb oil and organic pollutants [24]. In this work, the PP derived hydrophobic porous aerogel was prepared by a facile and environmentally-friendly method [26,27]. The textural features of the investigated materials were determined and systemically discussed by various characterization techniques (Fourier transform infrared (FTIR), Brunauer–Emmett–Teller (BET), X-ray diffraction (XRD), scanning electron microscope (SEM), TG). More importantly, the as-prepared PP-based aerogel showed remarkably high recyclability (greater than $40 \text{ g g}^{-1}$, for diesel oil) even after 10 cycles, which made it an attractive candidate for use in efficient oil spill clean-up and water purification.

# 2. Material and methods

## 2.1. Materials

PP was provided by local WalMart as solid waste. The yellow skin of PP was chipped off, then the sponge residue was thoroughly washed with deionized water to remove all the dirty particles. After

that, PP was grounded into granules after drying in an oven at 100°C for 10 h. The PP powder was stored in a glass desiccator for further use. Polyvinyl alcohol (PVA) and methyltrimethoxysilane (MTMS) were purchased from Aladdin Industrial Corporation. Ethanol, sodium hydroxide (NaOH), *N,N*-dimethylformamide (DMF) and dimethyl sulfoxide (DMSO) were purchased from the Kermel Chemical Reagent Co., Ltd (Tianjin, China). Diesel, crude oil and pump oil were supplied by local Sinopec Group (Dalian, China). Soya bean oil, peanut oil, sunflower oil etc. were purchased from a local market in Dalian, China. All chemicals were of analytical grade, and all chemicals were used without further purification.

## 2.2. Pomelo peel pretreatment

A certain amount of grounded PP powder (80–100 meshes) was immersed into a solution of 4 wt% sodium hydroxide with a solid–liquid ratio of 1 : 15 and the mixture was kept at 40°C for 6 h under magnetic stirring. After that, the product was filtered and thoroughly washed in deionized water until the filtrate was colourless and transparent (pH = 7). The obtained material was dried at 60°C for 24 h.

## 2.3. Preparation of pomelo peel-based sponge aerogels (PSA-0, PSA-1 and PSA-2)

Firstly, 1 g pretreated PP powder was added to the beaker containing 99 g H$_2$O. The resulting mixture was continuously stirred at room temperature (25 ± 1°C) for 0.5 h using a high shear dispersing emulsifier. Then, the suspension was frozen for 24 h under −18°C, and subsequently freeze-dried at a condenser temperature of −55°C under vacuum for 24 h to obtain sponge sample PSA-0. The aerogel PSA-1 and PSA-2 were prepared using the same method as PSA-0. The pretreated PP powder was mixed with filter paper or PVA with a mass ratio of 7 : 3 to prepare sample PSA-1 and PSA-2, respectively.

## 2.4. Preparation of hydrophobic sponge aerogels (HPSA-0, HPSA-1 and HPSA-2)

In detail, the aerogel PSA-0 was placed in a glass bottle. An amount of 1.5 ml MTMS and 1.5 ml deionized water were inserted into a beaker, respectively. After that, two small beakers were placed inside the above-mentioned 250 ml bottle. Then, the glass bottle was tightly sealed and placed in an oven at 80°C for 12 h. Finally, the silanized sample was placed in a vacuum oven at 60°C for 24 h to remove the unreacted silane. The achieved samples were labelled as HPSA-0. The HPSA-1 and HPSA-2 were prepared following the same procedure as with HPSA-0.

## 2.5. Density and porosity

The apparent density of aerogel (ρ) was calculated based on the following equation:

$$\rho = \frac{m}{(2\pi DH)/4},$$
(2.1)

where $\rho$ represents the density of aerogel (g cm$^{-3}$), $m$ is the quality of the aerogel (gram), $D$ is the diameter of cross section of aerogels (centimetre) and $H$ is the height of the aerogel (centimetre).

Aerogel porosity was calculated by the aerogel apparent density $\rho$ and skeleton density $\rho_0$ using the following equation [28,29]:

$$P = \left[1 - \left(\frac{\rho}{\rho_0}\right)\right] \times 100\%,$$
(2.2)

where $P$ is the porosity of aerogels (%), $\rho$ is the apparent density of aerogel (g cm$^{-3}$) and $\rho_0$ is the aerogel skeleton density (g cm$^{-3}$).

Based on data in the literature, we used the density of cellulose (1.500 g cm$^{-3}$) and PVA (1.267 g cm$^{-3}$) as the solid density [30], so, $\rho_{0(\text{HPSA-0})} = 1.500$ g cm$^{-3}$, $\rho_{0(\text{HPSA-1})} = 1.500$ g cm$^{-3}$ and $\rho_{0(\text{HPSA-2})} = 1.430$ g cm$^{-3}$ (the mass ratio of PP and PVA was 3 : 7).

## 2.6. Morphology characterization

The morphology of the samples was observed using a scanning electron microscope (SEM, JSM-6460LV) equipped with energy dispersive X-ray spectroscopy (EDS). The crystallinity of HPSA-0 (HPSA-1, HPSA-2) was analysed by X-ray diffraction (XRD, SmartLab from Rigaku Co.). Water contact angles (WCAS) were measured by a contact angle system (KSV CM20, Finland) under ambient conditions. Fourier transform infrared (FTIR) spectroscopy spectra were recorded using a Bruker Tensor 27 spectrometer. And $N_2$ adsorption–desorption experiments were carried out at 77 K using a SA 3100 apparature (Beckmann Coulter) to study the textural features of the samples (specific surface area, pore diameter/volume). Thermal gravimetric analysis was conducted on a Thermo plus TG-8120 apparatus. Typically, 50 mg of the aerogel samples (HPSA-0, HPSA-1 and HPSA-2) were heated from 20°C to 700°C at a rate of 10°C $min^{-1}$ under a $N_2$ flow 50 ml $min^{-1}$.

## 2.7. Oil and organic solvent absorption capacity

To assess the absorption capacity of the sponge aerogels, a series of oils and organic solvent were selected. An appropriate amount of oil or organic solvent was poured into a beaker, then the aerogel sample (initial weight was $m_0$) was immersed in oil. The sample was taken out at intervals of 5 s and the oil located at material surface was carefully scraped with filter paper. The sample weight ($m$) was recorded after the test. This process was repeated several times until the aerogel sample reached adsorption equilibrium. The oil absorption capacity $Q$ and oil absorption rate $V$ of aerogels were quantified by formulas 1.3 and 1.4, respectively [31–33].

$$Q = \frac{m - m_0}{m_0},$$ (2.3)

where $Q$, $m$ and $m_0$ represent the oil absorption capacity of the aerogel (g $g^{-1}$), the weight of aerogel after absorption (gram) and the weight of aerogel before absorption (gram).

$$V = \frac{Q}{t},$$ (2.4)

where $V$ represents the oil absorption rate of samples (g $s^{-1}$).

## 2.8. Reusability

In order to evaluate the reusability of the aerogels, 10 consecutive cycles of absorption and regeneration were carried out. The saturated sample was immersed into 200 ml of anhydrous ethanol for 0.5 h at 70°C. The oil dissolved in anhydrous ethanol completely, then it was separated from the obtained mixture by vacuum distillation. The anhydrous ethanol was recycled.

# 3. Results and discussion

## 3.1. Morphologies of the sponge aerogel

The morphologies of the sponge aerogels before and after silanization were observed via SEM images. As shown in figure 1a–c, the aerogel surface was flaked with a small number of interspaces, and the macropores and micropores made up with interconnected PP powder particles and filter paper fibres or PVA were clearly observed in the aerogel cross section. These interspaces greatly enhanced the adsorption efficiency of aerogels. [34,35] Compared with raw PP (electronic supplementary material, figure S1), the aerogels showed a multilevel and an interconnected three-dimensional pore structure which was interweaved by flexible sheets. Noticeably, the results suggested that a large number of the tunnel structures were distributed in three aerogel samples, which made it easy for oil to enter the aerogels.

In addition, the aerogels modified with MTMS (figure 1d,e) possessed an obviously layered structure, the surface shape was unchanged and the pore size was kept constant, which could reach up to 10–1000 μm. Unlike HPSA-1(PP/filter paper), a large amount of filamentous fibre structure in HPSA-2(PP/PVA) was determined. Furthermore, as a water-soluble linear polymer, PVA presented its adhesive ability between PP powder particles, leading to the aerogel surface smoother than HPSA-1. However, it still possessed a layered structure and multi-level pore structure.

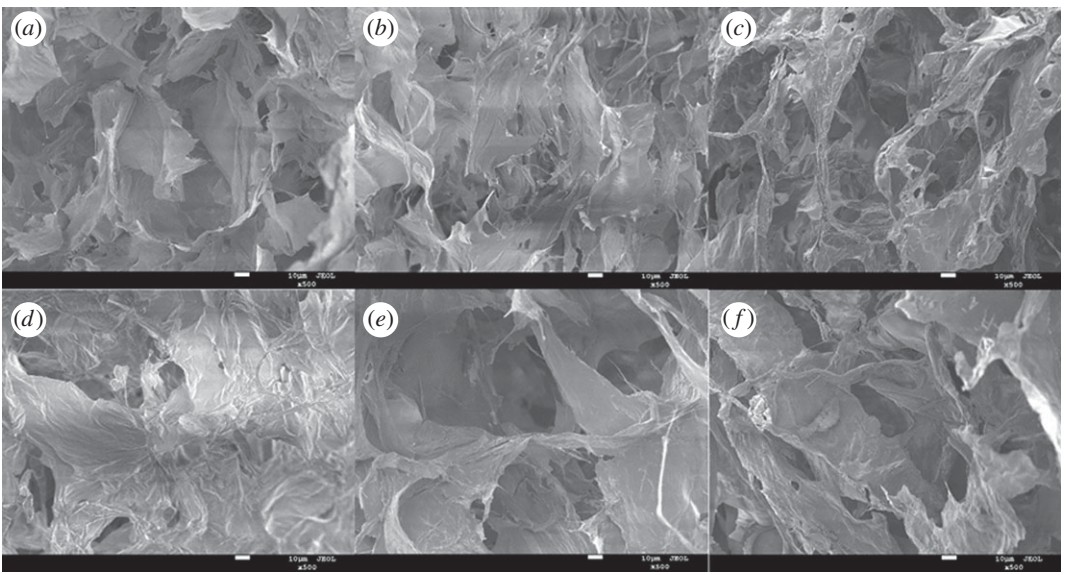

**Figure 1.** SEM images of: (*a*) PSA-0, (*b*) PSA-1, (*c*) PSA-2, (*d*) HPSA-0, (*e*) HPSA-1 and (*f*) HPSA-2.

**Table 1.** Density, porosity and specific surface areas of pomelo peel-based sponge aerogels.

| aerogel | $\rho$ (g cm$^{-3}$) | $\rho_0$ (g cm$^{-3}$) | porosity (%) | BET surface area (m$^2$ g$^{-1}$) |
|---|---|---|---|---|
| PSA-0 | 0.0306 | 1.500 | 98.0 | 11.6 |
| PSA-1 | 0.0214 | 1.500 | 98.6 | 16.0 |
| PSA-2 | 0.0204 | 1.430 | 98.6 | 7.0 |

The density, porosity and specific surface areas of PP-based sponge aerogels were listed in table 1. It revealed that the lowest density was *ca* 0.0204 g cm$^{-3}$ and its porosity could be as high as 98.6%. All aerogel samples exhibited ultralight and highly porous (approx. 98%) features. The specific surface area was measured using the BET method. Among three aerogel samples, the surface area of HSPA-1 was larger than that of the others, probably due to the loose and porous structure of filter paper fibre. This result indicated that filter paper fibre was favourable for the formation of the multi-stage pore structure of aerogel. On the other hand, the microstructure of aerogel could be adjusted by the different additives, which might present a facile method of controlling material structure [36].

Nitrogen adsorption–desorption tests have also been conducted to determine textural properties of the HPSA samples. Figure 2 shows the nitrogen adsorption–desorption isotherms for HPSA aerogels prepared with a different doping material. It could be observed that the HPSA-2 presented a small hysteresis loop at relative pressure from 0.4 to 0.6, indicating the mesoporous structure of the HPSA-2 sample. Similarly, the pore types of HPSA-0 samples were mainly mesoporous, which was in line with the results of pore size distribution curve (figure 2*d,e*).

## 3.2. Chemical component of aerogels

FTIR and EDX analysis were used to identify the surface chemical component of aerogel samples, and the results are shown in figure 3 and table 2, respectively. As shown in figure 3, the dominant peaks in the 34 271 036 and 902 cm$^{-1}$ were ascribed to the stretching vibrations of −OH, C−O−C [28] and β-glycosidic linkage, respectively. It indicated that the surfaces of the raw PP were rich in hydroxyl, carbonyl, carboxyl and aromatic groups [37]. Additionally, the bands at 750 ($\delta$ (C−H)) and 2980 cm$^{-1}$ ($\nu$ (C−H)) were assigned to the vibration's characteristic of the CH$_3$ in silane. Besides, Si−CH$_3$ bending vibration located at 1250–1390 cm$^{-1}$ was also noted. The characteristic peaks of Si−O−Si bonds in the siloxane were overlapped by the C−O bonds. The results revealed that MTMS could covalently form a strong cross-linked Si-O-Si monolayer coating under a modified reaction condition [38]. After silanized modification, the −OH groups were displaced by O-Si(CH$_3$)$_3$ groups from MTMS resulting in the hydrophobic property of the aerogels [29].

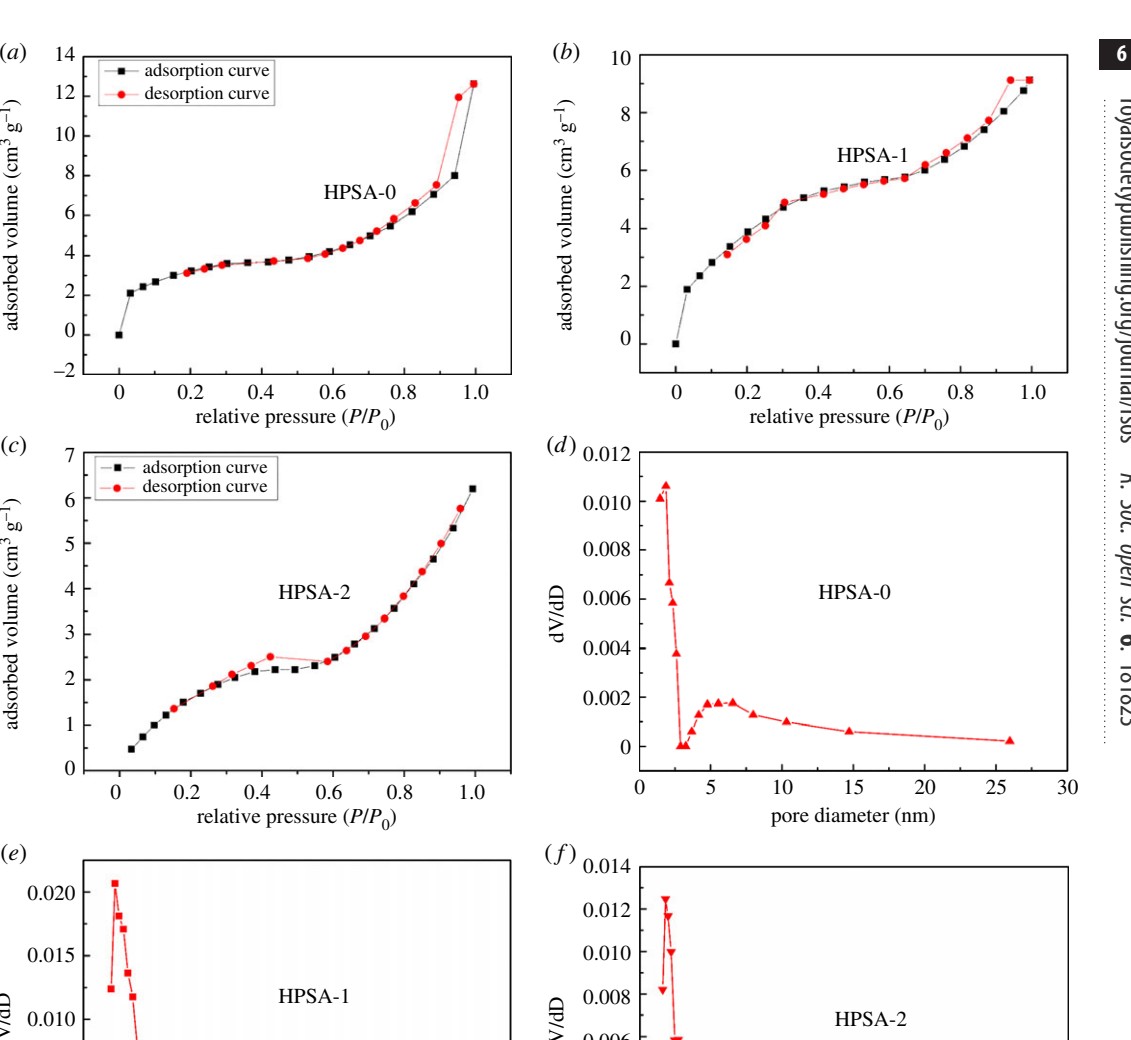

**Figure 2.** The $N_2$ adsorption–desorption isotherms of: (a) HPSA-0, (b) HPSA-1, (c) HPSA-2; the pore size distribution curves of: (d) HPSA-0, (e) HPSA-1 and (f) HPSA-2.

Table 2 shows the element component of raw PP, PSA-1, PSA-2 and silanized modified aerogel (HPSA-1, HPSA-2). In fact, there was no silicon element in the raw PP and alkali pretreatment PP, and silicon element was detected in the sponge aerogel (1.89 wt% and 1.02 wt%).

The above results showed that sponge aerogel samples were successfully modified by MTMS during the fabrication of hydrophobic aerogel.

It was known that crystallinity played an important role in the mechanical properties of aerogels. Figure 4 shows the XRD patterns of the modified aerogel samples (HPSA-0, HPSA-1 and HPSA-2) [30]. The results showed that all the samples presented a peak at around $2\theta = 23°$ (002 peak) [35]. It could be seen from the XRD pattern that the HPSA-0 and HPSA-1 had an intense diffraction peak at $2\theta = 22.60°$. This peak corresponded to the graphitic carbon structure of raw PP. The peak for HPSA-2 became narrow and stronger, suggesting that the crystallinity of HPSA-2 was higher than other samples.

## 3.3. Thermal properties of aerogels

Considering the heat insulation application, thermal stability of aerogels is of great importance [36]. The results of thermogravimetry (TGA) and derivative thermogravimetry (DTG) are shown in figure 5. It was

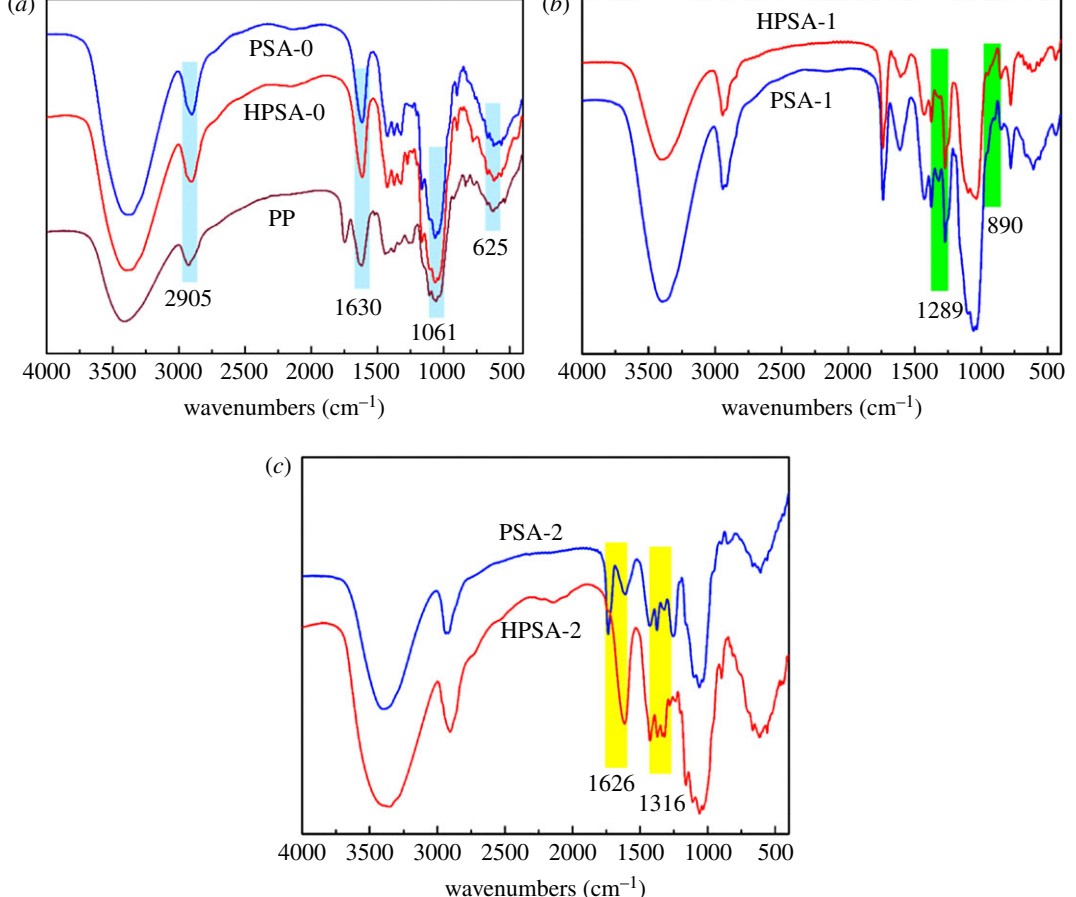

**Figure 3.** FTIR spectra of (*a*) PSA-0 and HPSA-0, (*b*) PSA-1 and HPSA-1, (*c*) PSA-2 and HPSA-2.

**Table 2.** EDS of PP, PP (pretreatment), PSA-1, HPSA-1, PSA-2 and HPSA-2.

| element | PP | PP (pretreatment) | PSA-1 | HPSA-1 | PSA-2 | HPSA-2 |
|---|---|---|---|---|---|---|
| C (wt%) | 63.87 | 62.82 | 54.99 | 57.37 | 47.8 | 51.44 |
| O (wt%) | 26.6 | 36.22 | 42.72 | 39.05 | 48.55 | 44.31 |
| S (wt%) | 0.44 | 0.48 | 0 | 0 | 0.21 | 0.2 |
| Si (wt%) | 0 | 0 | 0 | 1.89 | 0 | 1.02 |

clearly observed that only a little weight loss occurred in the range of $25-100°C$ owing to the evaporation of the adsorbed water. The main weight loss started at 227, 225 and 212°C for HPSA-0, HPSA-1 and HPSA-2, respectively in TGA. The degradation process reached its maximum at 313, 314 and 298°C, respectively as displayed in DTG. It is worth mentioning that the thermal stability of HPSA-0 was similar to HPSA-1. This result indicated that the higher temperature of thermal decomposition was attributed to the removal of lignin from the fibres and the addition of filter paper in aerogel. However, the thermal stability of HPSA-2 was lower in comparison with the others, probably because of the decomposition of PVA. According to the literature, linear PVA would be degraded and produce low molecular weight products when the temperature was greater than 264°C [39]. The thermal stability of PVA can be improved mainly by increasing its degree of alcoholysis, thereby the PVA with 99% alcoholysis was selected as the starting material.

## 3.4. Wettability

Desirable oil/water selectivity is a critical property for oil absorbents [40,41]. To determine the effect of silanized modification on the hydrophobicity of aerogel, WCA was measured to check the

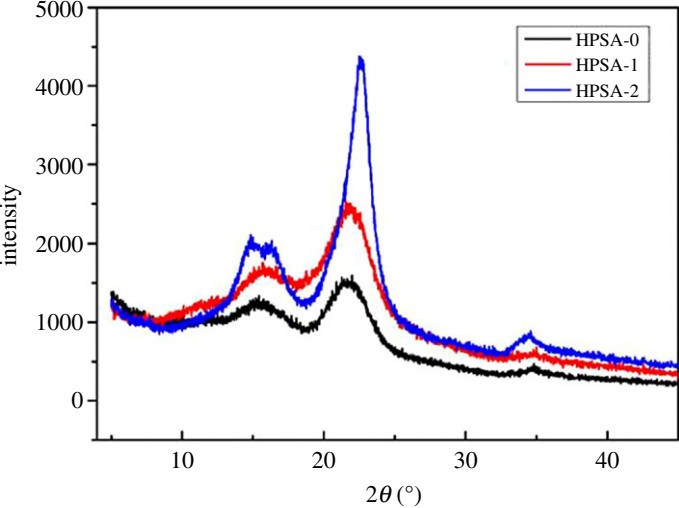

**Figure 4.** XRD patterns of HPSA-0, HPSA-1 and HPSA-2.

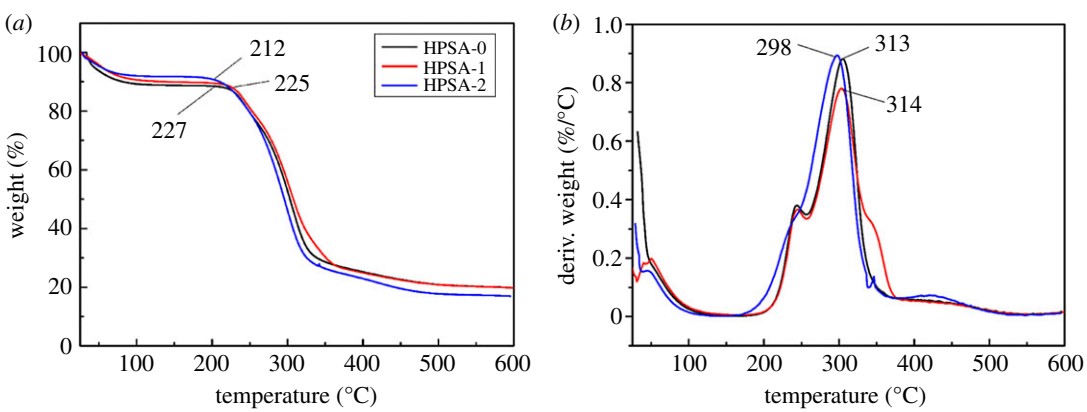

**Figure 5.** TGA (*a*) and DTG (*b*) thermograms of HPSA-0, HPSA-1 and HPSA-2.

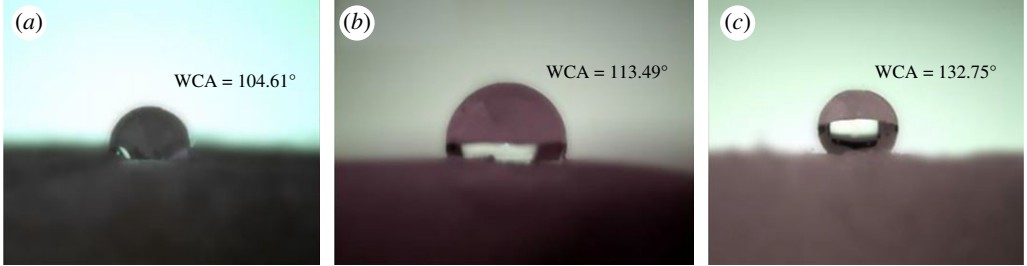

**Figure 6.** Water contact angle of the (*a*) HPSA-0, (*b*) HPSA-1 and (*c*) HPSA-2.

hydrophobicity. When the contact angle was higher than $90°$, the material was classified as hydrophobic [42]. As expected, the MTMS-modified aerogels became hydrophobic after silylation. Noticeably, water formed droplets on the aerogel surface and maintained its spherical shape with a contact angle of $132.75°$ (figure 6). This result could be ascribed to successful silanizing modification.

As shown in figure 7*a*, water (coloured with methylene blue) formed spherical droplets on the surface of HPSA-1, while kerosene (coloured with Sudan 3) was rapidly and completely absorbed. This revealed that the treatment of the sample with silanizing reagent showed the characteristics of hydrophobicity [43]. In figure 7*b*, the lumpy sponge aerogels suspended on the water surface also elucidated the hydrophobicity features of the modified sponge aerogels. The excellent hydrophobic performance of modified aerogels could be used in effective oil/water separation. MTMS-modified PP-based aerogel, due to its low density, high porosity and surface hydrophobicity, may be a promising candidate for the elimination of oils and organic pollutants.

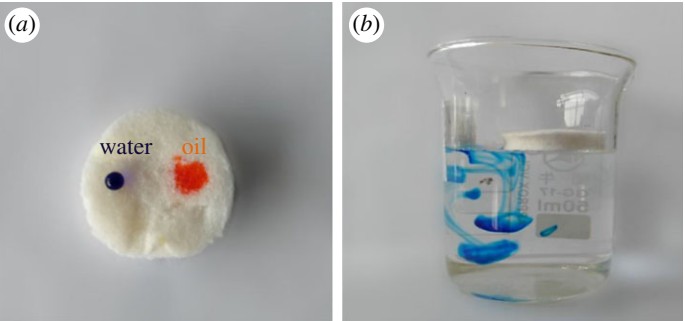

**Figure 7.** (*a*) Absorbed behaviour of the HPSA-1 (water and oil were coloured by methylene blue and Sudan 3, respectively), (*b*) behaviour of HPSA-2 in water.

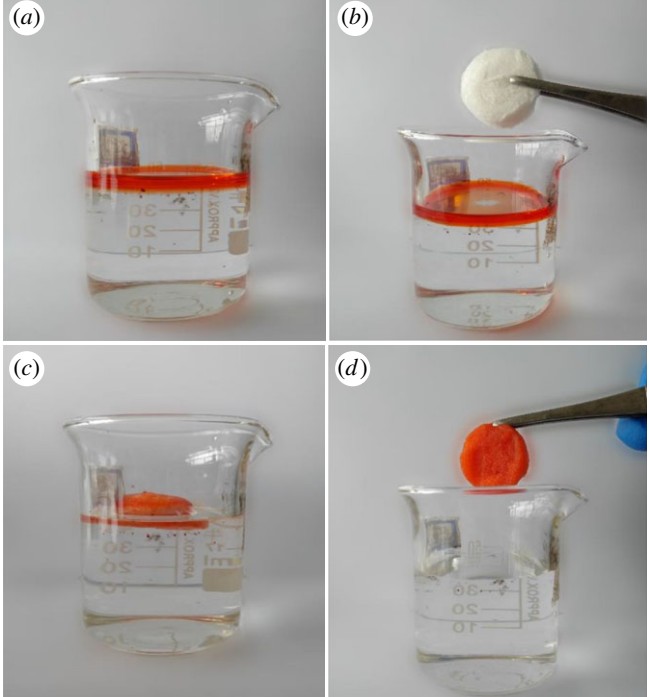

**Figure 8.** (*a*–*d*) Oil absorption process of the modified aerogel HPSA-2.

## 3.5. Absorption capacity

Oil leakages lead to serious pollution of the water environment, and the treatment of oil and organic pollutants from water has attracted considerable academic and commercial interest [44]. Sorption processes of aerogels occurred because of their micro and microporous structures, and also the inter- and intra-fibre interactions of the aerogel [45]. The PP-based aerogels possessed a hierarchical pore structure, as already discussed by SEM, BET analysis, so the aerogel can absorb oil rapidly. As shown in figure 8*a*–*d*, the kerosene (stained with Sudan 3 dye) was put into the beaker containing water, and then a modified sponge aerogel HPSA-2 was added into the beaker. The dyed kerosene was absorbed by the modified aerogel within 1 min, then the water in the beaker was clean. The result indicated that the modified samples possessed excellent adsorption capacity for oils and organic pollutants.

Figure 9*a*–*c* introduces the adsorption capacity of HPSA-0, HPSA-1 and HPSA-2 samples for some commonly used oils/organic solvents (crude oil, diesel oil, kerosene, engine oil, DMF, DMSO, ethanol, *n*-hexane etc.). The oil's adherence to the aerogel's surface happens mainly due to the intramolecular interaction and van der Waals forces [46]. The different sorption capacity was attributed to the density, the molecular dimension, the surface tension and hydrophobicity of the organic solvents and oils [41]. As could be clearly observed, the adsorption capacity of HPSA-1 and HPSA-2 for various oils/organic solvents was better than that of HPSA-0. It was ascribed to a satisfied network structure that was formed by filter paper fibres (PVA) and PP closed interaction. Furthermore, PVA will provide

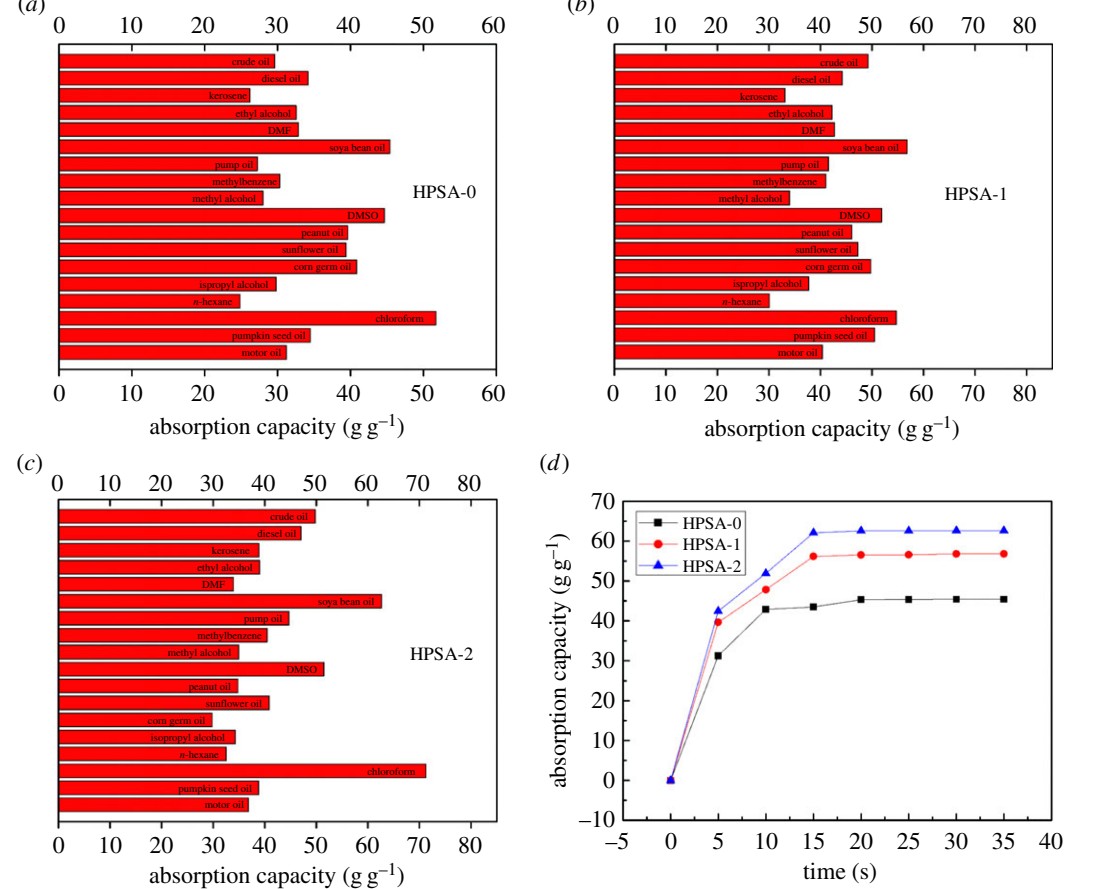

**Figure 9.** Absorption capacities for various oils and organic solvents (*a*) HPSA-0, (*b*) HPSA-1 and (*c*) HPSA-2. (*d*) Absorption rate curves of the HPSA-0, HPSA-1 and HPSA-2.

**Table 3.** The adsorption capacity and adsorption rate of sponge aerogels on soya bean oil.

| additive | mass ratios of additive-pomelo peel | absorption capacity (g g$^{-1}$) | absorption rate (g s$^{-1}$) |
|---|---|---|---|
| filter paper fibre | 2 : 8 | 53.4 | 3.56 |
| | 3 : 7 | 56.8 | 3.79 |
| | 6 : 4 | 52.4 | 3.49 |
| | 5 : 5 | 45.0 | 3.00 |
| PVA | 2 : 8 | 61.0 | 4.07 |
| | 3 : 7 | 62.6 | 4.18 |
| | 4 : 6 | 51.0 | 3.40 |
| | 5 : 5 | 51.1 | 3.41 |

good properties as toughness and flexibility for composite aerogel. Herein, the absorption capacity of HPSA-2 for chloroform and DMSO reached as high as 71.3 g g$^{-1}$, 51.5 g g$^{-1}$, respectively. In addition, the adsorption capacity of the HPSA-1 sample for pumpkin seed oil and crude oil was 50.5 g g$^{-1}$ and 49.2 g g$^{-1}$, respectively. In conclusion, the adsorption capacity of HPSA-1 and HPSA-2 for most oils and organic solvents was promising and attractive.

Figure 9*d* shows the absorption rate of HPSA-0, HPSA-1 and HPSA-2 samples for soya bean oil. In all cases, the adsorption rate of the three samples reaching saturation was below 20 s. Noticeably, the time for the HPSA-2 sample was as short as 15 s. This illustrated that the adsorption rate of the as-prepared sponge aerogel in this work was quite fast for oil phase and organic solvent. And both the adsorption capacity and adsorption rates greatly depended on the types of oils and organic solvents.

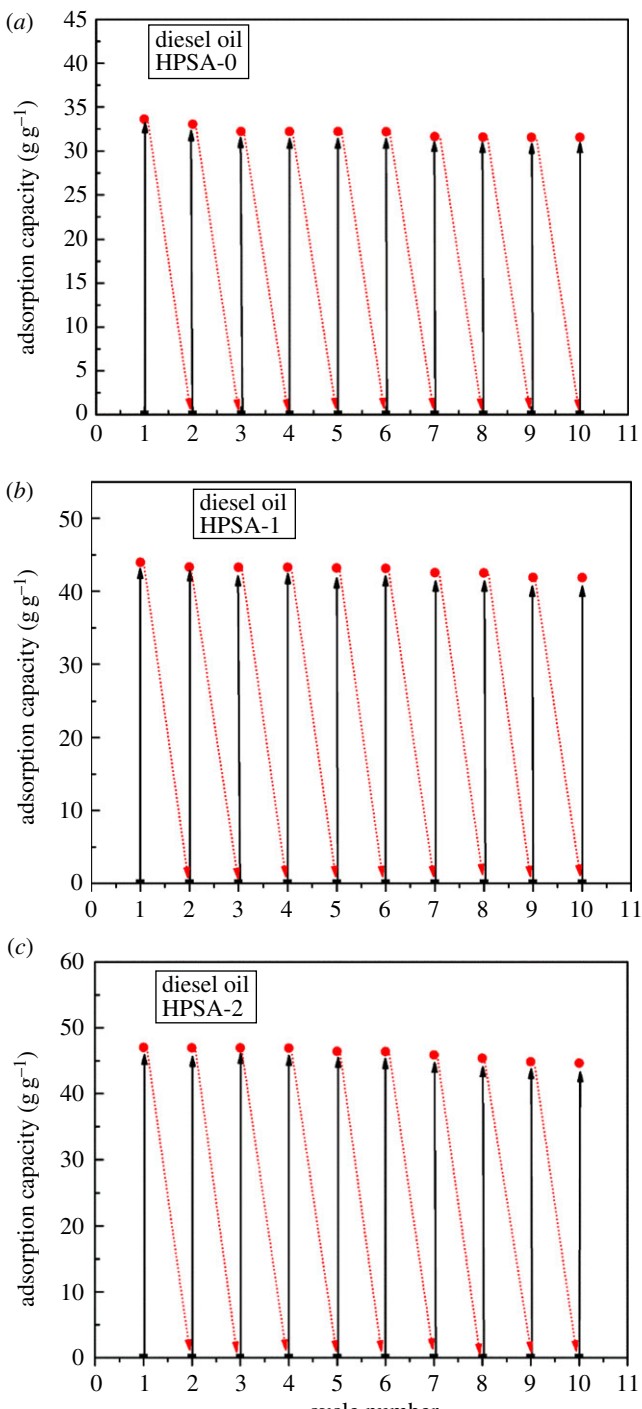

**Figure 10.** Reusability of the (*a*) HPSA-0; (*b*) HPSA-1; (*c*) HPSA-2.

As could be seen from table 3, the synthesized sponge aerogels presented the highest adsorption capacity and/or adsorption rate when the mixed mass ratio was 3:7 regardless of whether PVA or filter paper was used. Compared with the oil absorption capacity of HPSA-0, it was concluded that the filter paper fibre or PVA could significantly increase the aerogel absorption capacity. The additives were favourable to forming a more flexible three-dimensional porous structure.

## 3.6. Reusability

Reusability of the oil absorbent was a critical criterion for its potential pilot-scale application [47]. Here, diesel oil was selected as a model adsorbate to determine the circulating life of three aerogel samples. According to figure 10*a*−*c*, the adsorption capacity of HPSA-0 samples maintained at 93.82% after 10

cycles of absorption and regeneration, while the capacity of HPSA-1 and HPSA-2 samples remained 94.66% and 94.92%, respectively. As a result, the modified aerogels were highly worthy of being employed several times without considerable reduction in their efficiency, demonstrating their superior reusability. On the other hand, it was important to note that the recycling life for different oils and organic solvents varied. This was probably due to the various properties of the oil and organic solvents themselves.

# 4. Conclusion

To sum up, the sponge aerogels were synthesized from the waste PP using a simple and feasible method. The aerogels were characterized by FTIR, SEM, XRD and BET. The results indicated that HPSAs aerogels had an interconnected three-dimensional porous morphology. In order to render the ultralight and porous sponge aerogel hydrophobic, we exposed it to vapour phase deposition of MTMS. After modification, the contact angle at the water/sponge aerogel boundary remained high (WCA = 132.75°), indicating a strong hydrophobic character. In addition, the HPSAs aerogels exhibited a promising adsorption capacity (for chloroform, up to $70 \, \text{g g}^{-1}$), and the as-prepared sponge aerogels could be recycled 10 times, with the adsorption capacity maintained at 94.92%. It is worth mentioning that the main advantage of the investigated material was its simplicity, low cost and environmental-friendliness. This will play a positive role in the development of highly efficient adsorbents for different types of oils and organic solvents.

Data accessibility. The datasets supporting this article have been uploaded as part of the electronic supplementary material.

Authors' contributions. G.S., Y.Q., F.T. and Y.C. contributed the conception of the study and design of the experiments, G.S. performed the experiments and wrote the paper, Y.Q. and Y. L. contribute reagents and analysis the data, W.C. and F.T. revised the paper. All the authors read and approved the final manuscript.

Competing interests. The authors declare no competing interests.

Funding. This study was supported by Liaoning Province Ocean and Fishery office (grant no. 201405) and Dalian Science and Technology office (grant no. 2015B11NC078).

Acknowledgements. Instrument Analysis centre of Dalian Polytechnic University is gratefully acknowledged for all the equipment employed.

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
