## [Reviewer comments · Royal Society Open Science]

Review History

RSOS-181823.R0 (Original submission)

Review form: Reviewer 1 (Sheng Han)

Is the manuscript scientifically sound in its present form?

Yes

Are the interpretations and conclusions justified by the results?

Yes

Is the language acceptable?

Yes

Is it clear how to access all supporting data?

Yes

Do you have any ethical concerns with this paper?

No

Have you any concerns about statistical analyses in this paper?

No

Recommendation?

Accept with minor revision (please list in comments)

Comments to the Author(s)

"Controllable synthesis of pomelo peel-based aerogel and its application in adsorption of oil/organic pollutants", reports preparation of oil-absorbing aerogel from a mixture of pomelo peel and polyvinyl alcohol (or filter papers) with methylsilane treatment. Selective oil absorption was obtained from an aerogel made of cheap raw materials, and the materials was easily prepared and cost effective. The structure of the article was reasonable and the experimental results were discussed in detail and in good agreement with the physicochemical features of the as-prepared materials. Based on the results, I recommend the publication of this manuscript after a minor modification. Following is a list of a few to address.

1. First of all, this manuscript is required to have consult in terms of English.
2. Why are the pomelo peel and polyvinyl alcohol (or filter paper) mixed together?
3. The introduction of this manuscript need improve.

Review form: Reviewer 2 (Huie Liu)**Is the manuscript scientifically sound in its present form?**

No

Are the interpretations and conclusions justified by the results?

No

Is the language acceptable?

No

Is it clear how to access all supporting data?

Yes

Do you have any ethical concerns with this paper?

No

Have you any concerns about statistical analyses in this paper?

I do not feel qualified to assess the statistics

Recommendation?

Major revision is needed (please make suggestions in comments)

Comments to the Author(s)

The paper has shown some interesting results. But there are still some problems I suggest to solve.

1. The English language should be improved because some sentences cannot be understood clearly and there are some mistakes in the expression.
2. I suggest the authors characterize the raw material, pomelo peel, so as to make clear the composition of it.
3. For the Eq. 1,2, please tell us how you obtained the aerogel skeleton density.
4. Are chemical bonds formed during the formation of the aerogels of PSAs and HPSAs? I think

more characterization results are needed to show this.

5. The properties of pores are obtained through N₂ adsorption-desorption method and only micropores and mesopores can be analyzed. While I think that maybe macropores are very important for oil adsorption. So I suggest more characterization on the macropores should be provide.

6. For the description in Line 8~9, P. 5, "Clearly, the sample was completely absorbed by the dyed kerosene, otherwise the water in the beaker was not absorbed by the sample", please describe how can you determine that there is no water adsorbed.

7. Can you please give results about the strength of the material? And can you please give images for the aerogels after reusing for many times?

8. Anhydrous ethanol was used in the oil separation. Maybe you can used other solvents consider its future application.

Review form: Reviewer 3 (Ying Zhou)

Is the manuscript scientifically sound in its present form?

No

Are the interpretations and conclusions justified by the results?

No

Is the language acceptable?

No

Is it clear how to access all supporting data?

No

Do you have any ethical concerns with this paper?

No

Have you any concerns about statistical analyses in this paper?

No

Recommendation?

Reject

Comments to the Author(s)

The manuscript reported a pomelo peel-based aerogel for oil/organic pollutants adsorption. The authors used the natural pomelo peel as the starting material, however, the employment of fiber-contained nature materials as the precursors is not a new topic in this field. Apart from that, I didn't see any attractive innovations on both the synthesis and performance. The most disappointing thing is that the authors did not give any data analysis and academic discussion on their results. The whole manuscript is more like a experiment report.

In order to improve the current manuscript, I suggest the author place their focus on the following aspects:

First, improve the architecture of the paper, including the logic and the organization. First come up a problem, then look back the current researches in this field, finally put forward your work.....

Then, as for the detail, try to explain the formation of the aerogel and understand the function of each units in your material. Try to understand the mechanism of the hydrophobic property after coupling with MTMS. Discuss and compare the advantages and disadvantages of the your

aerogels prepared in this paper with other reported works on the synthesis, properties and oil/organic solvent adsorption capacity. Before doing this, I highly recommend the author to read more related literatures to make a clear understanding of the current research status.

Finally, please try your best to improve the english writing. I found many sentences that I even could not understand: "...Different from adding filter paper aerogel, the addition of PVA aerogels can see a large number of filamentous fiber structure, mainly due to the fiber in the filter paper is formed by PP fiber [27], the scattered distribution in a suspended solution.....The synthesized sponge aerogels in this work did not reach the superhydrophobic state of the samples after hydrophobic modification ($WCA \gg 150^\circ$) [33] as illustrated in the literature, which was regarded as the main reason for the amount of hydrophobic modification agents. It was anticipated to use a small amount of MTMS in order to modify the sample to achieve similar hydrophobic effect..."

Review form: Reviewer 4 (Yuchang Zhang)

Is the manuscript scientifically sound in its present form?

Yes

Are the interpretations and conclusions justified by the results?

Yes

Is the language acceptable?

Yes

Is it clear how to access all supporting data?

Yes

Do you have any ethical concerns with this paper?

No

Have you any concerns about statistical analyses in this paper?

No

Recommendation?

Accept with minor revision (please list in comments)

Comments to the Author(s)

In this manuscript the authors investigated oil adsorbent using renewable pomelo peel and filter paper (or polyvinyl alcohol) via facile method. This biomass-based aerogel shows low density, high porosity, hydrophobicity, high absorption capacity and excellent recyclability. Based on these results, this paper is valuable. I recommend accepting this paper after minor revised.

Following is a list of a few to address:

1. In page 1 line 45: author wrote "in situburning", it should be "in-situ burning". similarly, in page 2 line 3, "nonocoating" should be "nanocoating".
2. In page 3, discuss of aerogel microstructure (SEM) should be improved.
3. The title of figure 5 need to correct.
4. The English level of manuscript should be updated by a native English editor.

Decision letter (RSOS-181823.R0)

23-Nov-2018

Dear Professor Tan:

Title: Controllable synthesis of pomelo peel-based aerogel and its application in adsorption of oil/organic pollutants
Manuscript ID: RSOS-181823

The editor assigned to your manuscript has now received comments from reviewers. We would like you to revise your paper in accordance with the referee and Subject Editor suggestions which can be found below (not including confidential reports to the Editor). Please note this decision does not guarantee eventual acceptance.

Please submit your revised paper before 16-Dec-2018. Please note that the revision deadline will expire at 00.00am on this date. If we do not hear from you within this time then it will be assumed that the paper has been withdrawn. In exceptional circumstances, extensions may be possible if agreed with the Editorial Office in advance. We do not allow multiple rounds of revision so we urge you to make every effort to fully address all of the comments at this stage. If deemed necessary by the Editors, your manuscript will be sent back to one or more of the original reviewers for assessment. If the original reviewers are not available we may invite new reviewers.

Please also include the following statements alongside the other end statements. As we cannot publish your manuscript without these end statements included, if you feel that a given heading is not relevant to your paper, please nevertheless include the heading and explicitly state that it is not relevant to your work.

- Ethics statement

Please clarify whether you received ethical approval from a local ethics committee to carry out your study. If so please include details of this, including the name of the committee that gave consent in a Research Ethics section after your main text. Please also clarify whether you received informed consent for the participants to participate in the study and state this in your Research Ethics section.

OR

Please clarify whether you obtained the necessary licences and approvals from your institutional animal ethics committee before conducting your research. Please provide details of these licences and approvals in an Animal Ethics section after your main text.

OR

Please clarify whether you obtained the appropriate permissions and licences to conduct the fieldwork detailed in your study. Please provide details of these in your methods section.

RSC Associate Editor:
Comments to the Author:
(There are no comments.)

RSC Subject Editor:
Comments to the Author:
(There are no comments.)

Reviewers' Comments to Author:
Reviewer: 1

Comments to the Author(s)

"Controllable synthesis of pomelo peel-based aerogel and its application in adsorption of oil/organic pollutants", reports preparation of oil-absorbing aerogel from a mixture of pomelo peel and polyvinyl alcohol (or filter papers) with methylsilane treatment. Selective oil absorption was obtained from an aerogel made of cheap raw materials, and the materials was easily prepared and cost effective. The structure of the article was reasonable and the experimental results were discussed in detail and in good agreement with the physicochemical features of the as-prepared materials. Based on the results, I recommend the publication of this manuscript after a minor modification. Following is a list of a few to address.

1. First of all, this manuscript is required to have consult in terms of English.
2. Why are the pomelo peel and polyvinyl alcohol (or filter paper) mixed together?
3. The introduction of this manuscript need improve.

Reviewer: 2

Comments to the Author(s)

The paper has shown some interesting results. But there are still some problems I suggest to solve.

1. The English language should be improved because some sentences cannot be understood clearly and there are some mistakes in the expression.
2. I suggest the authors characterize the raw material, pomelo peel, so as to make clear the composition of it.
3. For the Eq. 1.2, please tell us how you obtained the aerogel skeleton density.
4. Are chemical bonds formed during the formation of the aerogels of PSAs and HPSAs? I think more characterization results are needed to show this.
5. The properties of pores are obtained through N₂ adsorption-desorption method and only micropores and mesopores can be analyzed. While I think that maybe macropores are very important for oil adsorption. So I suggest more characterization on the macropores should be provide.
6. For the description in Line 8~9, P. 5, "Clearly, the sample was completely absorbed by the dyed kerosene, otherwise the water in the beaker was not absorbed by the sample", please describe how can you determine that there is no water adsorbed.
7. Can you please give results about the strength of the material? And can you please give images for the aerogels after reusing for many times?
8. Anhydrous ethanol was used in the oil separation. Maybe you can used other solvents consider its future application.

Reviewer: 3

Comments to the Author(s)

The manuscript reported a pomelo peel-based aerogel for oil/organic pollutants adsorption. The authors used the natural pomelo peel as the starting material, however, the employment of fiber-contained nature materials as the precursors is not a new topic in this field. Apart from that, I didn't see any attractive innovations on both the synthesis and performance. The most disappointing thing is that the authors did not give any data analysis and academic discussion on their results. The whole manuscript is more like a experiment report.

In order to improve the current manuscript, I suggest the author place their focus on the following aspects:

First, improve the architecture of the paper, including the logic and the organization. First come up a problem, then look back the current researches in this field, finally put forward your work.....

Then, as for the detail, try to explain the formation of the aerogel and understand the function of each units in your material. Try to understand the mechanism of the hydrophobic property after coupling with MTMS. Discuss and compare the advantages and disadvantages of the your aerogels prepared in this paper with other reported works on the synthesis, properties and oil/organic solvent adsorption capacity. Before doing this, I highly recommend the author to read more related literatures to make a clear understanding of the current research status.

Finally, please try your best to improve the english writing. I found many sentences that I even could not understand: "...Different from adding filter paper aerogel, the addition of PVA aerogels can see a large number of filamentous fiber structure, mainly due to the fiber in the filter paper is formed by PP fiber [27], the scattered distribution in a suspended solution.....The synthesized sponge aerogels in this work did not reach the superhydrophobic state of the samples after hydrophobic modification ($WCA \gg 150^\circ$) [33] as illustrated in the literature, which

was regarded as the main reason for the amount of hydrophobic modification agents. It was anticipated to use a small amount of MTMS in order to modify the sample to achieve similar hydrophobic effect...”

Reviewer: 4

Comments to the Author(s)

In this manuscript the authors investigated oil adsorbent using renewable pomelo peel and filter paper (or polyvinyl alcohol) via facile method. This biomass-based aerogel shows low density, high porosity, hydrophobicity, high absorption capacity and excellent recyclability. Based on these results, this paper is valuable. I recommend accepting this paper after minor revised.

Following is a list of a few to address:

1. In page 1 line 45: author wrote "in situburning", it should be "in-situ burning". similarly, in page 2 line 3, "nonocoating" should be "nanocoating".
2. In page 3, discuss of aerogel microstructure (SEM) should be improved.
3. The title of figure 5 need to correct.
4. The English level of manuscript should be updated by a native English editor.

Author's Response to Decision Letter for (RSOS-181823.R0)

See Appendix A.

Decision letter (RSOS-181823.R1)

03-Jan-2019

Dear Professor Tan:

Title: Controllable synthesis of pomelo peel-based aerogel and its application in adsorption of oil/organic pollutants

Manuscript ID: RSOS-181823.R1

It is a pleasure to accept your manuscript in its current form for publication in Royal Society Open Science. The chemistry content of Royal Society Open Science is published in collaboration with the Royal Society of Chemistry.

RSC Associate Editor
Comments to the Author:
(There are no comments.)

Reviewer(s)' Comments to Author:

Appendix A

Dear Editor and Reviewers

Please find the enclosed **REVISED** version of our manuscript:

Ref. No.:RSOS-181823

“Controllable synthesis of pomelo peel-based aerogel and its application in adsorption of oil/organic pollutants” **and the Response to the reviewers’ REMARKS**

We sincerely thank the Reviewers and the editor for their valuable comments, which have certainly helped us to further improve the paper. The changes introduced are marked in red color in the new version.

Reviewers' Comments to Author:

Referee: 1

COMMENTS TO THE AUTHOR:

Reviewer #1: The manuscript (RSOS-181823), "Controllable synthesis of pomelo peel-based aerogel and its application in adsorption of oil/organic pollutants", reports preparation of oil-absorbing aerogel from a mixture of pomelo peel and polyvinyl alcohol (or filter papers) with methylsilane treatment. Selective oil absorption was obtained from an aerogel made of cheap raw materials, and the materials were easily prepared and cost effective. The structure of the article was reasonable and the experimental results were discussed in detail and in good agreement with the physicochemical features of the as-prepared materials. Based on the results, I recommend the publication of this manuscript after a minor modification. Following is a list of a few to address.

1. First of all, this manuscript is required to have consult in terms of English.

R: We have made a deep correction according to the Reviewer's comments. In the new version, many grammatical or type errors have been revised in order to make it more reasonable and meaningful for the readers.

2. Why are the pomelo peel and polyvinyl alcohol (or filter paper) mixed together?

R: We are particularly grateful to the reviewer to help us further improve the manuscript. Lignocellulosic biomasses that are one of the important sources of cellulose have recently attracted a great deal of attention as starting materials for aerogel preparation. But many researches focused on nanocellulose or cellulose crystals as a raw material to prepare aerogel. There will bring additional cost or/and complex process to obtain aerogel. In our work, a facile and eco-friendly process was used to prepare aerogel and the raw materials were pomelo peel and filter paper (PVA). The main component of filter paper was cellulose fiber, and long and soft cellulose fibers might be used as scaffold in aerogel. The flexible aerogel was obtained due to a satisfied network structure that formed by cellulose fibers and pomelo peel closed interaction. PVA is a water-soluble polymer, it is biodegradable and provide good properties as toughness and flexibility at relatively low cost.

3. The introduction of this manuscript need improve.

R: We are particularly grateful to the reviewer to help us further improve the manuscript. We have re-written the instruction according to the Reviewer's suggestion.

Reviewer #2: The paper has shown some interesting results. But there are still some problems I suggest to solve.

1. The English language should be improved because some sentences cannot be understood clearly and there are some mistakes in the expression.

R: Thanks for the reviewer's kindly suggestion. We have revised the paper carefully. Many grammatical or typographical errors, and incomprehensible sentences were corrected in order to make this paper more meaningful and reasonable.

2. I suggest the authors characterize the raw material, pomelo peel, so as to make clear the composition of it.

R: We greatly agreed with the suggestion of the reviewer. In the revised version, we have characterized the starting material pomelo peel by FTIR and EDS. The results were shown in Table 3 and Figure 3(a).

3. For the Eq. 1.2, please tell us how you obtained the aerogel skeleton density.

R: The density of the PVA was 1.27 g.cm^{-3} , according to the manufacturer's data sheet. Meanwhile, the density of cellulose fiber (pretreatment PP and filter paper fiber) was 1.50 g.cm^{-3} based on previous literature data. We can calculate the aerogel skeleton density as following:

Aerogel skeleton density (g.cm^{-3}) = $1.5 \times \text{cellulose fiber percent content} + 1.27 \times \text{PVA percent content}$

4. Are chemical bonds formed during the formation of the aerogels of PSAs and HPSAs? I think more characterization results are needed to show this.

R: During the formation of the aerogels of PSAs and HPSAs, there were new chemical bond Si-O formed. In this paper, the FTIR and EDS were used to characterized the samples. The results were shown in Figure 3 and Table 2.

5. The properties of pores are obtained through N₂ adsorption-desorption method and only micropores and mesopores can be analyzed. While I think that maybe macropores are very important for oil adsorption. So I suggest more characterization on the macropores should be provide.

R: We completely agreed with the suggestion of the reviewer. In this paper, we used SEM technique to indicate macropores of aerogel sample. In accordance with the SEM images, aerogels possessed a hierarchical pore structure that led to higher absorption capacity.

6. For the description in Line 8~9, P. 5, “Clearly, the sample was completely absorbed by the dyed kerosene, otherwise the water in the beaker was not absorbed by the sample”, please describe how can you determine that there is no water adsorbed.

R: We have repeated the experiment. In detail, we put the aerogel immerse into the water for 20S, then measured the weight of aerogels. The photograph was presented in support information, the result revealed that the aerogel could absorbed a little water, the absorption capacity was 0.33g/g. So, we have corrected this part.

7. Can you please give results about the strength of the material? And can you please give images for the aerogels after reusing for many times?

R: Considering the reviewer’s suggestion, we have investigated the strength of the aerogel HPSA-1. As shown in Figure S2, the aerogel exhibited a good mechanic strength. The aerogel could maintain its shape for a long time under a 200g weight. The images for the aerogels after reusing for 1, 5 and 10 times were shown in Figure S4. After 10 cycles, only a little change was observed for the surface morphology of the aerogel sample, indicating that the structure of aerogel didn’t destroy in the process of recycle.

8. Anhydrous ethanol was used in the oil separation. Maybe you can used other solvents consider its future application.

R: We are particularly grateful to the reviewer suggestion. In our follow-up study, we will try to use other solvents to removal the oil in aerogel.

In addition, I recommend this manuscript is required to have consult in terms of English.

R: Thanks a lot for the suggestion of the reviewer, the spelling and syntax errors have been checked and corrected carefully.

Reviewer:3The manuscript reported a pomelo peel-based aerogel for oil/organic pollutants adsorption. The authors used the natural pomelo peel as the starting material, however, the employment of fiber-contained nature materials as the precursors is not a new topic in this field. Apart from that, I didn't see any attractive innovations on both the synthesis and performance. The most disappointing thing is that the authors did not give any data analysis and academic discussion on their results. The whole manuscript is more like a experiment report.

In order to improve the current manuscript, I suggest the author place their focus on the following aspects:

1-First, improve the architecture of the paper, including the logic and the organization. First come up a problem, then look back the current researches in this field, finally put forward your work...

Then, as for the detail, try to explain the formation of the aerogel and understand the function of each units in your material. Try to understand the mechanism of the hydrophobic property after coupling with MTMS. Discuss and compare the advantages and disadvantages of the your aerogels prepared in this paper with other reported works on the synthesis, properties and oil/organic solvent adsorption capacity. Before doing this, I highly recommend the author to read more related literatures to make a clear understanding of the current research status.

R: In the new manuscript, we added more than forty relevant literatures to exhibit and support our experiment results and discussion. Especially, we have re-written the instruction according to the Reviewer's suggestion, and try to compare the advantages and disadvantages of our aerogels with other reported works.

2-Finally, please try your best to improve the english writing. I found many sentences that I even could not understand: "...Different from

adding filter paper aerogel, the addition of PVA aerogels can see a large number of filamentous fiber structure, mainly due to the fiber in the filter paper is formed by PP fiber [27], the scattered distribution in a suspended solution.....The synthesized sponge aerogels in this work did not reach the superhydrophobic state of the samples after hydrophobic modification ($WCA \gg 150^\circ$) [33] as illustrated in the literature, which was regarded as the main reason for the amount of hydrophobic modification agents. It was anticipated to use a small amount of MTMS in order to modify the sample to achieve similar hydrophobic effect...

R: We have revised this paper carefully. Many typos, grammatical errors, and incomprehensible sentences were corrected in order to make this paper more meaningful and reasonable.

Reviewer: 4 Comments to the Author(s) In this manuscript the authors investigated oil adsorbent using renewable pomelo peel and filter paper (or polyvinyl alcohol) via facile method. This biomass-based aerogel shows low density, high porosity, hydrophobicity, high absorption capacity and excellent recyclability. Based on these results, this paper is valuable. I recommend accepting this paper after minor revised. Following is a list of a few to address:

1. In page 1 line 45: author wrote "in situburning", it should be "in-situ burning". similarly, in page 2 line 3, "nonocoating" should be "nanocoating".

R: We are particularly grateful to the reviewer to point out our error in expressing. We have corrected the words from "in situburning" and "nonocoating" to "in-situ burning" and "nanocoating", respectively.

2. In page 3, discuss of aerogel microstructure (SEM) should be improved.

R: In the revised version, we have re-written this part according to the reviewer's comments.

3. The title of figure 5 need to correct.

R: The title of Figure 5 was correct in revised paper.

4. The English level of manuscript should be updated by a native English editor.

R: In line with the suggestion of the reviewer, both the spelling and syntax errors have been checked and corrected carefully.